# Deep Eutectic Solvents Based on Carboxylic Acids and Glycerol or Propylene Glycol as Green Media for Extraction of Bioactive Substances from *Chamaenerion angustifolium* (L.) Scop.

**DOI:** 10.3390/molecules28196978

**Published:** 2023-10-08

**Authors:** Alena Koigerova, Alevtina Gosteva, Artemiy Samarov, Nikita Tsvetov

**Affiliations:** 1Laboratory of Medical and Biological Technologies, Federal Research Centre “Kola Science Centre of the Russian Academy of Sciences”, Fersmana Str. 14, Apatity 184209, Russia; a.koygerova@ksc.ru; 2Tananaev Institute of Chemistry and Technology of Rare Elements and Mineral Raw Materials—Subdivision of the Federal Research Centre «Kola Science Centre of the Russian Academy of Sciences», Akademgorodok 26a, Apatity 184209, Russia; angosteva@list.ru; 3Department of Chemical Thermodynamics and Kinetics, Institute of Chemistry, Saint Petersburg State University, Universitetskiy Prosp. 26, St. Petersburg 198504, Russia; samarov@yandex.ru

**Keywords:** *Chamaenerion angustifolium* (L.) Scop., deep eutectic solvents, extraction

## Abstract

*Chamaenerion angustifolium* (L.) Scop. is one of the promising sources of biologically active compounds and a valuable industrial crop. Recently, green extraction methods have become more topical. One of them is the application of deep eutectic solvents (DESs). The aim of this work was the synthesis and characterization of DES consisting of glycerin or propylene glycol with malonic, malic, or citric acids, evaluation of their effectiveness for extracting useful substances from *C. angustifolium* during ultrasonic extraction, description of kinetics, and optimization of extraction conditions. DESs were obtained and characterized with FTIR. Their effectiveness in the process of ultrasound-assisted extraction of biologically active substances from *C. angustifolium* was estimated. Kinetic parameters describing the dependence of the total phenolic, flavonoids, and antioxidant content, free radical scavenging of DPPH, and concentration of flavonoid aglycons (myricetin, quercetin, and kaempferol) via time in the range of 5–60 min at 45 °C are obtained. Extraction conditions were optimized with the Box–Behnken design of experiment. The results of this work make it possible to expand the scope of DES applications and serve the development of *C. angustifolium* processing methods.

## 1. Introduction

In recent years, the attention of scientists has been increasingly focused on the study of the useful potential of wild plant raw materials characteristic of northern latitudes [1,2,3]. It is known that depending on the region of growth, the same plant species can accumulate different amounts of biologically active substances [4]. This is due to many factors, including the adaptive ability of plants to adapt to climate change. Thus, plants growing in northern latitudes can be promising raw materials for the production of biologically active compounds.

One of the most promising and valuable industrial crops growing in the northern latitudes is *Chamaenerion angustifolium* (L.) Scop., or fireweed. It is a perennial herbaceous plant of the Onagraceae family, growing in sparse woodlands, in clearings, in dry sandy places, and highly mineralized soils, with the absence of a fertile layer [5]. It is used in traditional medicine and herbal medicine [6]. Among the biologically active compounds contained in the aboveground parts of *C. angustifolium* are flavonoids, tannins, phenolic carboxylic acids, as well as a large amount of vitamin C. Extracts of this plant have anti-inflammatory, antifungal, antiproliferative, antiandrogenic, antioxidant, and anti-cancer properties [7]. In addition, it is a valuable summer honey plant. There are data on the high antimicrobial activity of honey from fireweed against *Streptococcus pneumoniae*, *S. pyogenes*, *Staphylococcus aureus*, and methicillin-resistant *S. aureus* [8]. It is known that diluted decoctions in the form of tea or low doses of dry extracts of fermented leaves can be used in specialized nutrition to increase the antioxidant status and adaptive potential as well as enhance immunity and body resistance to stress and other damaging factors for people working in complicated environmental conditions, for example in the Arctic [9].

In recent years, the development of green methods of biologically active component extraction has become increasingly relevant—in particular, the use of deep eutectic solvents (DESs) [10,11,12,13]. They are binary or triple mixtures, the components of which are bound by strong hydrogen bonds and, as a result, have a reduced melting point below or close to room temperature [14]. A variety of combinations of DES and plant material compositions open up the widest prospects for research both from a fundamental and applied point of view. There are many applications of DESs as solvents for reactions [15], energy applications [16], applications in analytical chemistry [17], electrochemistry [18], and biomass treatment [19,20]. One of the most interesting applications of DES is extraction carried out with the use of additional exposure to ultrasound, microwave radiation, etc. [21]. This additionally expands the number of alternative directions of scientific and technological developments.

DESs may consist of “natural” organic components, such as acids and amino acids, sugars, vitamins, etc. [22], forming natural deep eutectic solvents (NADESs). DESs are divided into five main types, and NADESs in some cases can be classified as Type V (a combination of a donor (HBD) and a hydrogen bond acceptor (HBA)) [23]. It is often difficult to confirm experimentally the formation of a particular eutectic mixture (having a minimum melting point), and we can talk just about “low melting mixtures”. However, following the majority of authors, in the future, the term “deep eutectic solutions”, (D)ESs, will be used in the work. At the same time, the formation of specific hydrogen bonds should be confirmed using FTIR [24].

Mixtures of dicarboxylic acids with sugars can be attributed to such combinations [25]. The presence of hydroxyl and carboxyl groups allows the formation of sufficiently strong hydrogen bonds. The same type of interaction can be characteristic of mixtures of polybasic organic acids with simpler polyols: for example, ethylene glycol, propylene glycol, and glycerol. Some works reported carboxylic acids and polyols as components of DES of type V [26,27]. Glycerol and propylene glycol are used as solvents for plant extract production, and thus, they can be considered promising components of (D)ESs [28,29,30,31].

(D)ESs have rarely been used for the extraction of biologically active substances from *C. angustifolium*. Previously, it was shown that (D)ESs based on choline chloride and polybasic organic acids have greater extraction efficiency than traditional solvents such as water and ethanol [32].

The purpose of this work was the synthesis and characterization of (D)ES consisting of glycerol or propylene glycol with malonic, malic, or citric acids, evaluation of their effectiveness for extracting useful substances from *C. angustifolium* during ultrasonic extraction, description of kinetics and optimization of extraction conditions.

## 2. Results and Discussion

The (D)ES structure was confirmed using the ATR-FTIR method. IR spectra are shown in Figure 1a–f and in Table 1. The most intense vibration bands of the carboxyl group are in italics.

There is no OH-group in MA. Mal and CA contain an OH-group, the vibrations of which are observed at about 3443 and 3495 cm^−1^, respectively. These peaks disappear when Mal and CA are mixed with alcohols and (D)ES is formed.

Mal and MA are anhydrous, while citric acid contains one water molecule. In the IR spectrum, the PG vibrational band at 3300 cm^−1^ refers to the O-H stretching of hydroxyl groups. A similar broad peak with a minimum of 3308 cm^−1^ is seen in the GL spectrum. The shift of these minima to the region of higher wavelengths occurs during the formation of the (D)ESs. This indicates the simultaneous interaction of all three components—carboxylic acid, polyhydric alcohol, and water. Therefore, the formation of new hydrogen bonds is indicative.

No fluctuations are observed in the region of 1600–1800 cm^−1^ in the spectrum of PG. A weak peak is present at 1646 cm^−1^ for GL. In turn, one C=O stretching vibration at 1693 cm^−1^ is observed for MA. Two vibrations each are present in the spectra of Mal and CA: 1737 cm^−1^ and 1678 cm^−1^ for Mal and 1741 cm^−1^ and 1685 cm^−1^ for CA. Moreover, peaks that have a higher wavelength have a lower intensity. All the studied (D)ESs have two peaks in this region; the first one is more intense. When comparing the more intense peaks, it can be seen that there is also a shift to the region of higher wavelengths. The increased intensity of bands associated with carbonyl stretching may indicate the formation of hydrogen bonding interactions, given that the formation of (D)ES is characterized by intermolecular interactions such as hydrogen bonding, Van der Waals forces, and electrostatic interactions [24].

Figure 1e demonstrates the effect of the amount of additional water on the structure of the resulting (D)ESs. The general patterns discussed above for three water contents (10, 15, and 20 molar parts) are preserved. However, the greatest difference in the intensities of the first and second peaks of the carboxyl group is observed for (D)ESs with 10 molar parts of water (1600–1750 cm^−1^). There is also a slight shift in the frequency of these peaks in the direction of decreasing wavelengths as the water content increases. A similar pattern is observed for the minimum of a wide peak in the region of 3000–3700 cm^−1^—the absorption region of OH groups. Consequently, there is a weakening of hydrogen bonds.

The results of these experimentally measured properties of density and viscosity of (D)ESs under study (with 10 molar parts of water) at 30 °C are presented in Appendix A.

It can be seen that the highest density and viscosity are observed for CAGL, while MAPG has the lowest density and MAGL has the lowest viscosity. In the pair of (D)ESs with the same acid, a higher density was obtained for glycerol. For the viscosity, there is no dependence on polyol type. The highest viscosities were obtained for (D)ESs based on citric acid. The viscosities of (D)ESs can affect the extraction rate due to diffusion limitations, so it is important to compare the kinetics of the processes for each (D)ES.

The kinetics of extraction of polyphenols and flavonoids in a medium of various (D)ES is well described by second-order reaction equations, and the 1/Yt vs. t dependences for all (D)ES are described by linear equations with *R*^2^ > 0.99 (Appendix A). The obtained parameters of the kinetic equations are presented in Appendix A. It can be noted that the fastest polyphenols are extracted using MAGL (k = 17.0 × 10^−3^ g/mg×min), while the slowest extraction process occurs in the case of MalGL (k = 0.6 × 10^−3^ g/mg×min). Flavonoids are extracted faster by MAPG (k = 65.6 × 10^−3^ g/mg×min) and slower when using CAGL (k = 4.4 × 10^−3^ g/mg×min), which may be due to the high density and viscosity. The lack of correlation between the kinetic parameters for TPC and TFC can be explained by the difference in the behavior of different groups of substances in different (D)ES and, accordingly, the different rates of their extraction. In our previous study [32], the highest rate constant for polyphenols was observed for ethanol, but its value was just 2.5 × 10^−3^ g/mg×min. Thus, the extraction of polyphenols is faster for MAGL than ethanol. The extraction rate for flavonoids in this work is also higher than that for ethanol in the work [32].

The highest yield of polyphenols is obtained for (D)ES CAGL, and then relatively close TPC values are obtained for MAGL and CAPG. Flavonoids are extracted better when using MAGL, while CAGL and CAPG have a statistically insignificant difference in efficiency (Figure 2a,b). It should be mentioned that in this work, the TPC and TFC values (300–350 mg GAE/g and 100–120 mg RE/g, respectively) are greater than the ones reached in previous work (250–300 mg GAE/g and 60–80 mg RE/g, respectively) [32].

The dependence of these characteristics of the extracts obtained on time, in general, is satisfactorily described by the kinetic equation, and for most (D)ESs, the linear approximation of 1/*Yt* vs. *t* is characterized by *R*^2^ > 0.99 (Appendix A). The kinetic parameters of TAC and DPPH are presented in Appendix A. However, in the case of CAGL, it was not possible to describe the change in DPPH over time using the equation used, since there is a decrease in the indicator after 20 min of extraction. It is also impossible to talk about a satisfactory description of TAC and DPPH for MalPG. This indicates more complex processes associated with the simultaneous extraction, decomposition, and/or evaporation from the liquid phase of substances with antiradical and antioxidant activity in these cases. In general, DPPH is characterized by kinetic dependencies that differ from other parameters of extracts [32,33]. The highest rate of change in TAC is observed for CAPG, and DPPH is observed for MalGL.

Extracts based on CAGL have the greatest antioxidant activity. However, according to the DPPH parameter, both (D)ES compositions with citric acid exhibit the lowest activity, while MAGL has the greatest free radical scavenging (Figure 2c,d).

The dependence of the yield of flavonoids, which are glycosides of myricetin, quercetin, and kaempferol, on time is described fairly accurately by the second-order reaction equation (Appendix A); however, for quercetin, strictly speaking, a different picture is observed: a gradual decrease in concentration during extraction in the first 20 min is obtained (Appendix A). After this time, the concentration of quercetin glycosides reaches a plateau. Kaempferol glycosides are completely extracted within 20–40 min, while the concentration of myricetin glycosides does not reach the equilibrium value within 60 min of extraction. However, in general, based on the data on the kinetics of the extraction of the sum of flavonoids, it can be assumed that the main process is completed in 40 min.

All glycosides of myricetin, quercetin, and kaempferol are best extracted using MAGL and CAPG. The extraction is worst for (D)ESs with malic acid (Figure 2e–g). It should be noted that in a pair of MalGL and MalPG, quercetin and kaempferol are better extracted to MalGL, while myricetin is better extracted to MalPG.

Based on the obtained data on extraction kinetics and comparison of extraction efficiency, (D)ES CAGL was selected for further work on the optimization of extraction conditions. (D)ESs based on citric acid also show higher effectiveness in the previous work [32].

The results of BBD optimization (Figure 3) are described quite well by polynomials
Y = a_0_ + a_1_A + a_2_B + a_3_C + a_4_AB + a_5_AC + a_6_BC + a_7_A^2^ + a_8_B^2^ + a_9_C^2^ + a_10_A^2^B + a_11_A^2^C + a_12_AB^2^

For all parameters, the most suitable models and their parameters were selected; in each case, lack of fit is not significant: the *R*^2^ for TPC was 0.914, TFC—0.995, TAC—0.959, concentration of myricetin—0.973, quercetin—0.988, kaempferol—0.992. The numerical values of model coefficients are given in Table 2. From the shape of the surfaces and the *p*- values for specific terms of the model, it can be seen that the water content in (D)ESs has a very weak effect on the yield of the final product, while the influence of temperature and volume/mass ratio is significant.

From the obtained equations, the optimal extraction conditions were estimated as follows: for TPC, TFC, and TAC, the optimal temperature is 60 °C, volume/mass ratio—24, and moles of water in (D)ES—20. For the aglycones of flavonoids, the optimal temperature is 55 °C, volume/mass ratio—10, and moles of water in (D)ES—20. In these conditions, the following yields may be obtained: TPC—212 mg GAE/g, TFC—74 mg RE/g, TAC—33 mg AAE/g, c(myricetin)—157 μg/mL, c(quercetin)—143 μg/mL, c(kaempferol)—53 μg/mL.

Here, we offer a workflow for a qualitative assessment of biologically active substances obtained by the extraction of (D)ESs based on glycerol and propylene glycol from a promising plant. In addition, this workflow can be extended to all types of plant extracts from fruits, flowers, leaves and roots. Glycols are a viable alternative to other reference solvents, such as ethanol and methanol, for the production of plant extracts with wide application in the medical, food, cosmetic and agricultural industries [28].

Glycerol is a widely used, non-toxic ingredient in cosmetic products. However, due to its high viscosity, it is not possible to use it undiluted for the preparation of plant extracts [30].

## 3. Materials and Methods

### 3.1. Plant Material

Leaves of *C. angustifolium* were collected on the experimental site of the Polar Alpine Botanical Garden Institute (67°34′ N 33°24′ E) during the flowering vegetation period in mid-August 2020. An average sample of aerial parts of plants was obtained from an area of 5 × 5 m. The drying and storage of plant material was following the recommendations from [34]. For extraction, plant material was milled with a household grinder and sieved through a sieve with 0.5 mm holes. The milled plant material was additionally dried at 45 °C until a constant mass was reached. The humidity of the dry sample was 0.134%.

### 3.2. Chemicals and Reagents

Malonic, malic, tartaric and citric acids, glycerol and propylene glycol (>99%, Vekton, St. Petersburg, Russia) were used for (D)ES preparation. 2,2-diphenyl-1-picrylhydrazyl (99%, Sigma-Aldrich, Burlington, MA, USA), Folin–Ciocalteu reagent (2M, Sigma-Aldrich), ammonium molybdate, potassium dihydrogenphosphate, aluminum chloride (>99%, Vekton, St. Petersburg, Russia), concentrated sulfuric and hydrochloric acids (>94%, Nevareactiv, St. Petersburg, Russia), gallic acid (98% Sigma-Aldrich, Burlington, MA, USA), rutin (≥94%, Sigma-Aldrich, Burlington, MA, USA), ascorbic acid (>99.7%, Hugestone, Nanjing, China), and ethanol (96%, RFK Company, St. Petersburg, Russia) were used for chemical analysis. Myricetin, quercetin and kaempferol analytical standards (>98%, Sigma-Aldrich), acetonitrile HPLC grade (Component Reactive, Moscow, Russia), glacial acetic acid (Vekton, St.Petersburg, Russia), and deionized water were obtained with the water purification system “Millipore Element” (Millipore, Burlington, MA, USA) and used for LC-UV analysis.

### 3.3. Deep Eutectic Solvents Preparation

The preparation of (D)ESs was carried out by mixing acid mixtures with glycols, and then the mixtures were heated at 60 °C for 3 h. Various combinations of components were tested, and ratios were selected that gave a homogeneous mixture (Table 3). To lower the (D)ES viscosity, water was added to them. To compare the efficiency of extraction using different (D)ESs, a water additive of 10 molar parts was used. To optimize extraction conditions using the Box–Behnken design of experiment, the amount of water varied.

In all experiments, freshly prepared (D)ESs were used to minimize the effect of the possible formation of esters, which is well known for (D)ES based on choline chloride and carboxylic acids [35].

### 3.4. Characterization of (D)ESs

IR spectra were recorded on a Nicolet 6700 FT-IR spectrophotometer (Thermo Fisher Scientific Inc., Hillsboro, OR, USA, 2010) in the 4000–550 cm^−1^ region (diamond ATR, 16 scans, resolution 4).

There are plenty of experimental data for (D)ESs formed by choline chloride and glycerol or ethylene glycol, and choline chloride and organic acids. However, data on (D)ESs formed by organic acids and glycerol or propylene glycol are not found in the literature. In this work, we measured the density of the (D)ESs under study using a DMA 5000 M density meter (Anton Paar GmbH, Graz, Austria); the measurement uncertainty is 0.00001 g·cm^−3^. Also for the (D)ESs, the viscosity was determined using a Modular Compact Rheometer MCR 702 (Anton Paar GmbH, Austria); the measurement uncertainty is 0.08 mPa·s.

### 3.5. Ultrasound-Assisted Extraction, Kinetics and Box–Behnken Design Optimization

Extraction was performed in the thermostated ultrasound bath Vilitek VBS 3-DP (Vilitek, Moscow, Russia, 2018) with an ultrasound power of 120 W and a frequency of 40 kHz.

To determine the optimal extraction time and select the most suitable extractant, extraction was carried out for 5–60 min at 45 °C with a volume/mass ratio of 20. The change in concentrations of target groups of components or flavonoids in the form of their aglycones was described by the second-order reaction kinetic equation, following the work [32]. From the coefficients of the linear equation for dependence, 1/Y_t_ vs. t (where Y_t_—yield of a target compound or group of compounds at time t), equilibrium yield (Y^eq^), and the rate constant (k) were calculated.

After selecting the most suitable (D)ES composition and extraction time, the extraction conditions, such as temperature, volume/mass ratio and water content in (D)ES were optimized. A Box–Behnken design of experiment (BBD) combined with response surface methodology (RSM) was applied for the optimal condition calculation. In this design of experiment, three parameters of extraction conditions have three levels. The temperature was in the range of 30–60 °C, the volume/mass ratio was in the range of 10–30, and the water content in (D)ESs was in the range of 10–30 molar parts. The combinations of parameters used in this work are presented in Table 4.

### 3.6. Chemical Analysis of Extracts

The total phenolic content (TPC) was measured using the Folin–Ciocalteu method, and the total flavonoid content (TFC) was measured using the reaction of complexation with aluminum chloride in a 90% (*v*/*v*) ethanol–water mixture. TPC is expressed as mg/g of gallic acid equivalent (GAE) per one gram of plant weight. TFC is expressed as mg/g of rutin equivalent (RE) per one gram of plant weight. All the methods are described in detail in [33].

### 3.7. Antioxidant Activity Measuring

The total antioxidant content (TAC) was measured using the phosphomolybdate method and was expressed as mg/g of ascorbic acid equivalent (AAE) per one gram of plant weight. Free radical scavenging was measured using the DPPH method and was expressed as % of inhibition in comparison with blank solution. Here, 95% ethanol was used as a solvent for the DPPH solution. The incubation was performed for 30 min at 25 °C in the dark. All these methods are also described in detail in [33].

### 3.8. LC-UV Analysis for Quantification of Aglycons of Extracted Flavonoids

For the quantitative determination of flavonoid aglycones, the acid hydrolysis described in [24] was performed with modifications. First, 100 μL of freshly prepared ascorbic acid solution in 4 M hydrochloric acid with a concentration of 1 mg/mL was mixed with 100 μL of row plant extract. The obtained mixture was incubated at 70 °C for 1 h in the plastic test tubes with screw caps. Then, the mixture was dissolved 5 times with 95% ethanol and was centrifugated for 10 min at 4 rpm in the laboratory centrifuge MiniSpin (Eppendorf, Hamburg, Germany, 2018).

LC-UV analysis was performed with a liquid chromatograph Milichrom A-02 with a ProntoSil-120-5-C18 AQ column, 75 × 2 mm, with a particle size of 5 μm (Econova, Novosibirsk, Russia, 2022). The gradient elution was performed using 1% (*v*/*v*) acetic acid in water (A) and acetonitrile (B) with the following program: 0–5 min—0% B, 5–30 min 0–100% B, 30–32 min 100% B, 32–35 min 100–0% B, flow rate—100 μL/min, sample volume—2 μL. The wavelength of detection was 254 nm. Solutions of myricetin, quercetin, and kaempferol in the concentration range of 6.25–100 μg/mL were used for calibration.

### 3.9. Statistical Analysis

All the measurements were made three times for each analysis. Statistical comparison was performed using factorial analysis of variance (ANOVA) and post hoc Tukey’s HSD test at *p* ≤ 0.05. The calculations were performed using MS Excel 2010 (Microsoft, Redmond, WA, USA). DesignExpert 11 (Stat-Ease, Minneapolis, MN, USA) software was used for the Box–Behnken design and the UAE condition optimization using response surface methodology.

## 4. Conclusions

In the present work, for the first time, (D)ESs based on carboxylic acids (malonic, malic, and citric) with glycerol or propylene glycol were applied for the ultrasound-assisted extraction of biologically active substances from *Chamaenerion angustifolium* (L.) Scop.

The (D)ESs used in the work are characterized in detail using FTIR; data on density and viscosity are obtained. (D)ESs based on glycerol have relatively higher densities in combination with the same acid. Dependences of the total phenolic, flavonoids, and antioxidant content (TPC, TFC, and TAC), free radical scavenging of DPPH, and concentration of flavonoid aglycons (myricetin, quercetin, and kaempferol) via time in the range of 5–60 min at 45 °C were approximated using second-order reaction equations. The highest yield of the target components was achieved when using (D)ES—citric acid + glycerol. For this (D)ES, extraction conditions were optimized with the Box–Behnken design of experiment, and the optimal conditions are the following: for the TPC, TFC, and TAC, the optimal temperature is 60 °C and the volume/mass ratio is 24, while for flavonoid aglycons, the optimal temperature is 55 °C and the volume/mass ratio is 10. The optimal molar content of water was 20. In these conditions, the following yields may be obtained: TPC—212 mg GAE/g, TFC—74 mg RE/g, TAC—33 mg AAE/g, c(myricetin)—157 μg/mL, c(quercetin)—143 μg/mL, and c(kaempferol)—53 μg/mL.

The results of this work make it possible to expand the scope of (D)ES application and serve the development of *C. angustifolium* processing methods within the framework of green chemistry technologies for the needs of the production of cosmetics, biologically active additives, and pharmaceuticals. In addition, a relevant issue remains regarding (D)ESs recycling, and this should be considered a further direction of research in (D)ESs application.

## Figures and Tables

**Figure 1 molecules-28-06978-f001:**
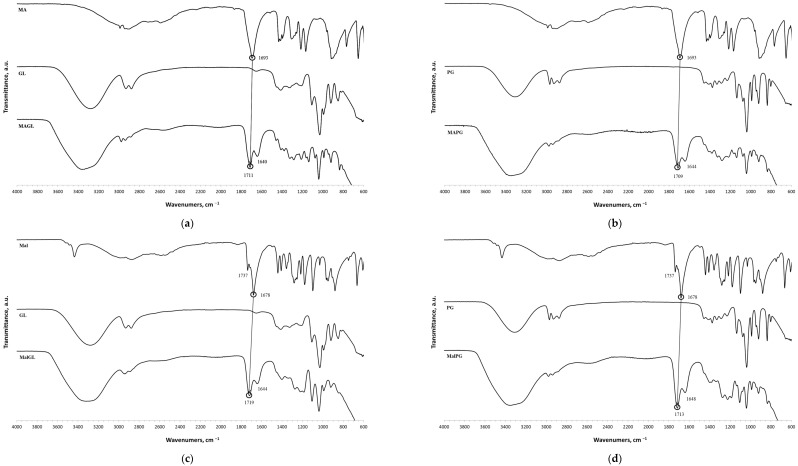
FTIR spectra of the initial components and the (D)ESs, used in the work: (**a**)—malonic acid + glycerol, (**b**)—malonic acid + propylene glycol, (**c**)—malic acid + glycerol, (**d**)—malic acid + propylene glycol, (**e**)—citric acid + glycerol, (**f**)—citric acid + propylene glycol.

**Figure 2 molecules-28-06978-f002:**
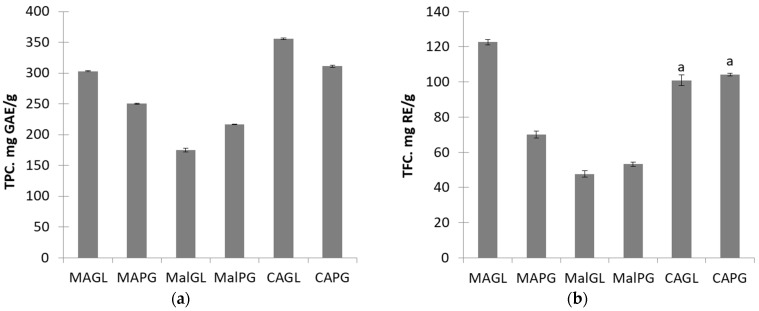
Comparison of total polyphenols (**a**) and total flavonoids (**b**) content, total antioxidant activity (**c**), free radical scavenging (**d**) of extracts, and content of glycosides of myricetin (**e**), quercetin (**f**), and kaempferol (**g**) in extracts based on various (D)ESs. The same letters denote the values, the difference between which is not statistically significant at *p* < 0.05.

**Figure 3 molecules-28-06978-f003:**
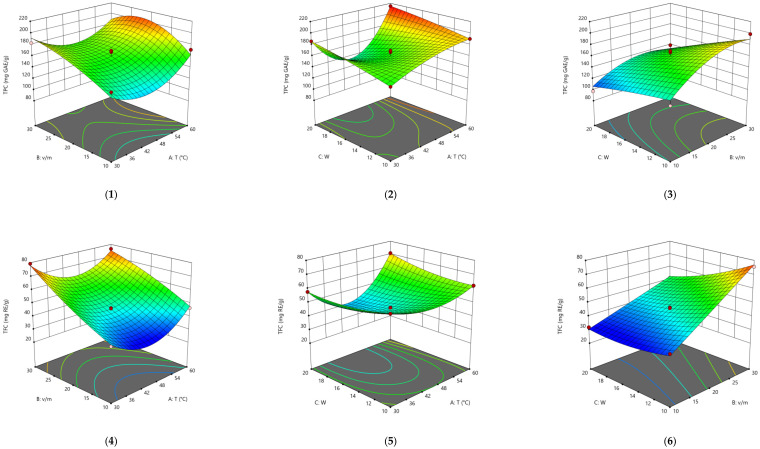
Response surfaces displaying the effect of extraction temperature (A), extraction time (B), and water content (C) on the extraction yield of TPC (**1**–**3**), TFC (**4**–**6**), TAC (**7**–**9**), myricetin (**10**–**12**), quercetin (**13**–**15**), and kaempferol (**16**–**18**).

**Table 1 molecules-28-06978-t001:** Characteristic bands of the initial components of the (D)ES and the (D)ES used in the work (the most intense peaks are highlighted in italics).

Component 1	Assignment, cm^−1^	Component 2	Assignment, cm^−1^	(D)ES + 10 H_2_O
MA	ν(C=O) 1693	Gly	ν(O-H) 3278	3358, *1711*, 1640
		PG	ν(O-H) 3300	3357, *1709*, 1644
Mal	ν(C=O) 1737, *1678*	Gly	ν(O-H) 3278	3297, *1719*, 1644
		PG	ν(O-H) 3300	3357, *1713*, 1648
CA	ν(C=O) 1741, *1685*	Gly	ν(O-H) 3278	3289, *1720*, 1644
		PG	ν(O-H) 3300	3360, *1711*, 1640

**Table 2 molecules-28-06978-t002:** Coefficients of models for BBD results fitting.

	Intercept	A	B	C	AB	AC	BC	A^2^	B^2^	C^2^	A^2^B	A^2^C	AB^2^
TPC	160.140	15.700	19.750	−22.125	−6.525	−1.400	−7.675	28.405	−17.970	−1.745		35.725	−10.575
*p*-values		0.090	0.014	0.032	0.423	0.859	0.352	0.011	0.057	0.820		0.020	0.363
TFC	45.800	3.225	15.925	−7.225	−2.275	3.225	−5.025	14.688	0.138	2.788		8.200	−2.400
*p*-values		0.016	<0.0001	0.001	0.053	0.016	0.003	<0.0001	0.882	0.025		0.001	0.119
TAC	16.100	2.013	3.463	0.200	2.775	2.650	−0.650	4.938	−2.213	2.113		3.100	
*p*-values		0.013	0.001	0.814	0.014	0.017	0.455	0.001	0.032	0.037		0.036	
Myricetin	178.420	4.708	−12.700	0.017	1.042	3.935	−0.747	−35.169	−21.229	2.908	13.489	6.482	
*p*-values		0.113	0.015	0.996	0.776	0.308	0.838	0.0001	0.002	0.429	0.040	0.244	
Quercetin	91.060	5.633	−24.525	−2.389	−2.954	5.350	−0.685	17.285	14.206	−16.995	−22.855	2.857	
*p*-values		0.047	0.001	0.468	0.377	0.139	0.831	0.002	0.005	0.002	0.003	0.537	
Kaempferol	32.991	2.330	−7.112	−1.141	−8.938	1.878	0.081	−0.404	−1.846	0.850	1.955		9.326
*p*-values		0.020	0.0002	0.068	<0.0001	0.043	0.912	0.577	0.042	0.265	0.104		0.0002

**Table 3 molecules-28-06978-t003:** Composition of (D)ES used in the work and their abbreviations.

Component 1	Component 2	Abbreviation	Molar Ratio Component 1: Component 2: Water
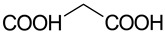 Malonic acid	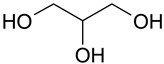 Glycerol	MAGL	1:1:10
	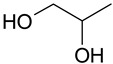 Propylene glycol	MAPG	1:2:10
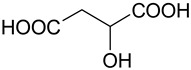 Malic acid	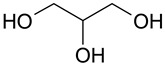 Glycerol	MalGL	1:2:10
	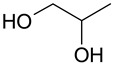 Propylene glycol	MalPG	1:2:10
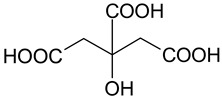 Citric acid	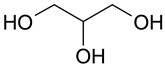 Glycerol	CAGL 10 H_2_O CAGL 15 H_2_O CAGL 20 H_2_O	1:4:10 1:4:15 1:4:20
	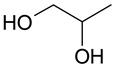 Propylene glycol	CAPG	1:4:10

**Table 4 molecules-28-06978-t004:** Parameters and their levels used in Box–Behnken design of experiment.

Temperature °C	Volume to Mass Ratio	Molar Parts of H_2_O
30 (−1)	20 (0)	5 (−1)
30 (−1)	10 (−1)	10 (0)
30 (−1)	30 (+1)	10 (0)
30 (−1)	20 (0)	15 (+1)
45 (0)	10 (−1)	5 (−1)
45 (0)	30 (+1)	5 (−1)
45 (0)	20 (0)	10 (0)
45 (0)	20(0)	10 (0)
45 (0)	20 (0)	10 (0)
45 (0)	20 (0)	10 (0)
45 (0)	20 (0)	10 (0)
45 (0)	10 (−1)	15 (+1)
45 (0)	30 (+1)	15 (+1)
60 (+1)	20 (0)	5 (−1)
60 (+1)	10 (−1)	10 (0)
60 (+1)	30 (+1)	10 (0)
60 (+1)	20 (0)	15 (+1)

## Data Availability

Not applicable.

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
