# Peer review of "Deep Eutectic Solvents Based on Carboxylic Acids and Glycerol or Propylene Glycol as Green Media for Extraction of Bioactive Substances from Chamaenerion angustifolium (L.) Scop."

_molecules, 2023, doi:10.3390/molecules28196978_

Round 1

Reviewer 1 Report

The manuscript entitled 'Deep eutectic solvents based on carboxylic acids and glycerol or propylene glycol as green media for extraction of bioactive substances from Chamanerion angustifolium (L.) Scop.' describes the preparation and use of non-ionic DES for the extraction of polyphenols. The DES used are original and the structure of the paper is scientifically rigorous. The accurate description of biomass cultivation is undoubtedly one of the strengths. However, there are some critical points that do not allow the publication of the paper in its present form.

Starting with the introduction, the part on DESs is very confusing and the applications in which DESs are used are not clear. DES are eutectic mixtures of two or more compounds that show a deviation from the ideal eutectic in terms of melting point (10.1007/s10953-018-0793-1). This difference may be due to interactions between HBA and HBD. In the case of DES or (D)ES reported by the authors, there is no HBA as neither polyols nor carboxylic acids are HBAs. The authors should therefore demonstrate that there is a deviation from the ideal eutectic in order to call them DES and alternatively call them eutectic or low melting mixtures. Furthermore, the authors should report many more applications of DESs as solvents for reactions (10.3390/molecules25030574), energy applications (10.3390/molecules27030709), applications in analytical chemistry (10.1016/j.microc.2017.07.015), electrochemistry (10.1016/j.electacta.2021.138189), biomass treatment (10.1039/d2gc03198a, 10.3390/molecules25163652). One of the most interesting is undoubtedly that of extractions, and therefore other references must also be added in combination with microwaves and ultrasound.

The most sensitive issue concerns the recycling of (D)ES solvents. The chemical instability of DES based on choline chloride and carboxylic acids is well known in the literature. Indeed, ester formation has been reported even at low temperatures (10.1021/acssuschemeng.9b01378). Solvent recycling is of paramount importance in the treatment of biomass. Therefore, the authors should check the stability of the solvent after treatment and carry out solvent recycling. To make the system environmentally and economically sustainable, this is mandatory. 

Minor editing of English language required

Author Response

We are grateful to the reviewer for all comments and recommendation.

Recommendation 1. DES are eutectic mixtures of two or more compounds that show a deviation from the ideal eutectic in terms of melting point (10.1007/s10953-018-0793-1). This difference may be due to interactions between HBA and HBD. In the case of DES or (D)ES reported by the authors, there is no HBA as neither polyols nor carboxylic acids are HBAs. The authors should therefore demonstrate that there is a deviation from the ideal eutectic in order to call them DES and alternatively call them eutectic or low melting mixtures.

Response: We are grateful to the reviewer for this comment. Indeed, for many combinations of HBS and HBA, very detailed studies are required to confirm the formation of a particular eutectic mixture. Determining the exact melting point of a mixture is often difficult due to the poor crystallization ability of DES. Of course, it is better to use the term “low melting mixtures” however, in this paper we adhere to the most common phrase in the literature “deep eutectic solvents”. The using of polyols and carboxylic acids is known in the literature (10.3390/molecules27030923 and 10.1039/c9gc00944b). More arguments have been added to the text of the manuscript.

Recommendation 2. Furthermore, the authors should report many more applications of DESs as solvents for reactions (10.3390/molecules25030574), energy applications (10.3390/molecules27030709), applications in analytical chemistry (10.1016/j.microc.2017.07.015), electrochemistry (10.1016/j.electacta.2021.138189), biomass treatment (10.1039/d2gc03198a, 10.3390/molecules25163652). One of the most interesting is undoubtedly that of extractions, and therefore other references must also be added in combination with microwaves and ultrasound.

Response: These examples of the use of DESs are undoubtedly important. We mentioned them in the paper.

Recommendation 3. The most sensitive issue concerns the recycling of (D)ES solvents. The chemical instability of DES based on choline chloride and carboxylic acids is well known in the literature. Indeed, ester formation has been reported even at low temperatures (10.1021/acssuschemeng.9b01378). Solvent recycling is of paramount importance in the treatment of biomass. Therefore, the authors should check the stability of the solvent after treatment and carry out solvent recycling. To make the system environmentally and economically sustainable, this is mandatory. 

Response: We thank the reviewer for pointing out very important and sensitive issues. Freshly prepared DES were used in the work, and the absence of significant formation of an ester bond was confirmed by FTIR. The issue of DES recycling is planned to be discussed in detail in our future works. Information about this has been added to the manuscript.

Reviewer 2 Report

I recommend this paper should be polished up focusing on:

·  highlight the purpose of the study in the Abstract

·  Introduction doesn't provide sufficient background and includes all relevant references. The research design is appropriate, but In Results and Discussion section are mostly results, with negligible discussion. The discussion should have this form: 1. the main results; 2. interpret why – mechanisms; 3. originality; 4. restrictions; 5. how it fits in with previous studies; 6. what can be further explored?

·  The conclusions are generally supported by the results, but the whole article has an insufficient emphasis on glycerol or 2 propylene glycol as green media for such a title.

·        Please give some arguments/reasons which make glycerol or propylene glycol as green media DES interesting for plant extractions or the herbal industry.

·        Explain briefly the Box-Behnken Design (BBD): What intention is followed? Without the knowledge of your previous work the reader have to understand what you are doing here.

·        Line 144: What kind of solvent did you use for the DPPH assay, and what incubation time and temperature?

·        English correction  (example: Milled plant material was additionally dried at 45°C to rich a constant mas... reach)

·        It is very unusual that you didn't compare results with your work Tsvetov, N.; Pasichnik, E.; Korovkina, A.; Gosteva, A. Extraction of Bioactive Components from Chamaenerion Angustifolium (L.) Scop. with Choline Chloride and Organic Acids Natural Deep Eutectic Solvents. Molecules 2022, 27, 376 4216, doi:10.3390/molecules27134216, which is almost the same principle, just with choline chloride. It is even on the front page in the Citation part forgotten from the last work.

Noticed:

·        In line 44 Add honey from fireweed… the high antimicrobial activity of honey from? against

·        In line 44 correct S. Аureus

·        Line 62 is incorrect, there are an abundance of papers. 62: However, there are relatively few papers devoted to synthesis and characterization such combinations of HBD and HBA [18,19].

 I don't recommend the paper for publication in the current status, but reconsider after the mentioned revisions.

English language needs grammar, spelling, punctuation, and typos correction.

Author Response

We are very grateful to the reviewer for all comments.

Recommendation 1· Highlight the purpose of the study in the Abstract

Response: Thank you for this recommendation. The purpose of the study was added to the Abstract.

Recommendation 2· Introduction doesn't provide sufficient background and includes all relevant references. The research design is appropriate, but In Results and Discussion section are mostly results, with negligible discussion. The discussion should have this form: 1. the main results; 2. interpret why – mechanisms; 3. originality; 4. restrictions; 5. how it fits in with previous studies; 6. what can be further explored? The conclusions are generally supported by the results, but the whole article has an insufficient emphasis on glycerol or 2 propylene glycol as green media for such a title.

Response: We add more information to the Introduction part about glycerol and propylene glycol. More information and comparison with previous results were added.

Recommendation 3·Please give some arguments/reasons which make glycerol or propylene glycol as green media DES interesting for plant extractions or the herbal industry.

Response:  We added information about this in the Introduction section.

Recommendation 4·Explain briefly the Box-Behnken Design (BBD): What intention is followed? Without the knowledge of your previous work the reader have to understand what you are doing here.

Response: We add explanation of BBD DOE.

Recommendation 5·        Line 144: What kind of solvent did you use for the DPPH assay, and what incubation time and temperature?

Response: We added information about this method.

Recommendation 6·English correction (example: Milled plant material was additionally dried at 45°C to rich a constant mas... reach)

Response: English correction was made

Recommendation 7·It is very unusual that you didn't compare results with your work Tsvetov, N.; Pasichnik, E.; Korovkina, A.; Gosteva, A. Extraction of Bioactive Components from Chamaenerion Angustifolium (L.) Scop. with Choline Chloride and Organic Acids Natural Deep Eutectic Solvents. Molecules 2022, 27, 376 4216, doi:10.3390/molecules27134216, which is almost the same principle, just with choline chloride. It is even on the front page in the Citation part forgotten from the last work.

Response: Thank you for this recommendation. We add a comparison of results.

Noticed:

  • In line 44 Add honey from fireweed… the high antimicrobial activity of honey from? against
  • In line 44 correct S. Аureus
  • Line 62 is incorrect, there are an abundance of papers. 62: However, there are relatively few papers devoted to synthesis and characterization such combinations of HBD and HBA [18,19].

Response:  All recommendations were taken into account, needed corrections were made.

Reviewer 3 Report

The paper 'Deep eutectic solvents based on carboxylic acids and glycerol or propylene glycol as green media for extraction of bioactive substances from Chamanerion angustifolium (L.) Scop' is interesting, well-documented. However, some minor corrections are required:

line 45 - aureus - lower case

line 63 - HBD and HBA - abbreviations should be explained

lines 109, 115, 184, 189, 190, 224 - upper case should be applied (cm-1, cm-3)

line 142, 148 - only reference [23] contains more detailed description of the applied method in submtted paper, it should be corrected

moreover, list of references should be corrected according to Instruction for Authors

no comments

Author Response

We are very grateful to the reviewer for all comments.

line 45 - aureus - lower case

line 63 - HBD and HBA - abbreviations should be explained

lines 109, 115, 184, 189, 190, 224 - upper case should be applied (cm-1, cm-3)

line 142, 148 - only reference [23] contains more detailed description of the applied method in submtted paper, it should be corrected

moreover, list of references should be corrected according to Instruction for Authors

Response: We are very grateful to the reviewer for his comments and suggestions. All recommendations are taken into account.

Round 2

Reviewer 1 Report

I would like to thank the authors for accepting the majority of my suggestions. However, one minor revision remains necessary. The presence of hydrogen bonding alone is not a justification for the formation of deep interactions in a eutectic mixture. Moreover, there is no hydrogen bond acceptor (HBA) in the eutectic mixture proposed by the authors. Both compounds are HBDs. Therefore, I suggest using the term (D)ES instead of DES throughout the discussion in the manuscript. The true 'deep' nature cannot be verified. Although mistakes have been made in this regard in the past, it is no longer possible to perpetuate the same errors in newly prepared systems.

Author Response

We are very grateful to the reviewer for valuable recommendations. Necessary corrections have been added to the manuscript.

Reviewer 2 Report

Recommendations were taken into account, but there are primarily results in the Results and Discussion section, with negligible discussion.

Author Response

We are very grateful to the reviewer for this recommendation. Additional text was added to Discussion section.